# *RNF135* Expression Marks Chemokine (C-C Motif) Ligand-Enriched Macrophage–Tumor Interactions in the Glioblastoma Microenvironment

**DOI:** 10.3390/cancers17193271

**Published:** 2025-10-09

**Authors:** Jianan Chen, Qiong Wu, Anders E. Berglund, Robert J. Macaulay, James J. Mulé, Arnold B. Etame

**Affiliations:** 1Department of Neuro-Oncology, H. Lee Moffitt Cancer Center and Research Institute, Tampa, FL 33612, USA; jianan.chen@moffitt.org (J.C.); qiong.wu@moffitt.org (Q.W.); 2Department of Quantitative Health Sciences, Division of Computational Biology, Mayo Clinic, 4500 San Pablo Road South, Jacksonville, FL 32224, USA; berglund.anders@mayo.edu; 3Departments of Anatomic Pathology, H. Lee Moffitt Cancer Center and Research Institute, 12902 Magnolia Drive, Tampa, FL 33612, USA; robert.macaulay@moffitt.org; 4Department of Immunology, H. Lee Moffitt Cancer Center and Research Institute, 12902 Magnolia Drive, Tampa, FL 33612, USA; james.mule@moffitt.org

**Keywords:** glioblastoma, *RNF135*, tumor-associated macrophages, single-cell RNA-seq, CCL3–CCR1

## Abstract

**Simple Summary:**

Glioblastoma remains lethal despite standard therapy. We identify that *RNF135*, an immune-related gene, marks a dominant subset of tumor-associated macrophages in GBM. Across TCGA/CGGA and single-cell datasets, *RNF135*-positive TAMs predict poorer survival and exhibit an immune state that is simultaneously activated and suppressive. Network analysis (CellChat) reveals that *RNF135* organizes chemokine-centric communication with aggressive tumor states, driven by the CCL3/CCL3L3–CCR1 axis, which strengthens TAM-to-TAM and TAM-to-tumor crosstalk. Spatial data localize *RNF135* to microvascular-proliferation regions. Drug-response modeling suggests that *RNF135*-high tumors are more sensitive to the MEK1/2 inhibitor selumetinib, consistent with MAPK-driven CCL signaling. Together, *RNF135* defines a macrophage program that promotes an inflamed yet immunosuppressive microenvironment and highlights the CCL–CCR1 axis and MEK inhibition as therapeutic avenues.

**Abstract:**

Background: Tumor-associated macrophages (TAMs) are essential regulators of the glioblastoma (GBM) microenvironment; their functional heterogeneity and interaction networks are not fully elucidated. We identify *RNF135* as a novel TAM-enriched gene associated with immune activation and adverse prognosis in GBM. Methods: To evaluate *RNF135*’s expression profile, prognostic significance, and functional pathways, extensive transcriptome analyses from TCGA and CGGA cohorts were conducted. The immunological landscape and cellular origin of *RNF135* were outlined using single-cell RNA-seq analyses and bulk RNA-seq immune deconvolution (MCP-counter, xCell and ssGSEA). Cell–cell communication networks between tumor cells and *RNF135*-positive and -negative tumor-associated macrophage subsets were mapped using CellChat. Results: *RNF135* predicted a poor overall survival and was markedly upregulated in GBM tissues. Functional enrichment analyses showed that increased cytokine signaling, interferon response, and innate immune activation were characteristics of *RNF135*-high samples. Immune infiltration profiling showed a strong correlation between the abundance of T cells and macrophages and *RNF135* expression. According to the single-cell analyses, *RNF135* was primarily expressed in TAMs, specifically in proliferation, phagocytic, and transitional subtypes. *RNF135*-positive TAMs demonstrated significantly improved intercellular communication with aggressive tumor subtypes in comparison to *RNF135*-negative TAMs. This was facilitated by upregulated signaling pathways such as MHC-II, CD39, ApoE, and most notably, the CCL signaling axis. The CCL3/CCL3L3–CCR1 ligand–receptor pair was identified as a major mechanistic driver of TAM–TAM crosstalk. High *RNF135* expression was also linked to greater sensitivity to Selumetinib, a selective MEK1/2 inhibitor that targets the MAPK/ERK pathway, according to drug sensitivity analysis. Conclusions: *RNF135* defines a TAM phenotype in GBM that is both immunologically active and immunosuppressive. This phenotype promotes inflammatory signaling and communication between cells in the tumor microenvironment. Targeting the CCL–CCR1 axis or combining *RNF135*-guided immunomodulation with certain inhibitors could be a promising therapeutic strategies for GBM.

## 1. Introduction

Glioblastoma (GBM) is the most aggressive and lethal primary malignant tumor of the adult central nervous system [1]. GBM grows quickly, spreads widely, and is resistant to treatment. The current standard of care for GBM is maximal safe surgical resection followed by radiotherapy and temozolomide chemotherapy [2]. Even with these different types of treatment, the average life expectancy is still only 14 to 16 months, and recurrence is almost certain [3,4]. The limited clinical efficacy is attributed not only to the genetic and phenotypic heterogeneity of tumor cells but also to the immunosuppressive tumor microenvironment (TME) in which they reside [5].

The immune system plays a critical role in tumor surveillance and elimination; however, the immunologically privileged status of the brain presents unique challenges to effective immunotherapy [6]. The blood–brain barrier (BBB) limits peripheral immune cell infiltration into the tumor microenvironment, while GBM cells further establish an immunosuppressive niche by downregulating antigen presentation, expressing immune checkpoint ligands, and recruiting regulatory T cells (Tregs) and tumor-associated macrophages (TAMs), thus promoting immune evasion and tumor progression [5,7]. TAMs represent the most abundant and functionally diverse immune cell population within the GBM microenvironment [8]. Single-cell RNA sequencing studies have revealed that TAMs comprise distinct functional subtypes—such as proliferation, antigen-presenting, regulatory, and hypoxic phenotypes—that contribute to immunoregulation, antigen processing, and suppression of antitumor responses [9,10]. These cells are increasingly recognized as key drivers of tumor progression. However, the molecular mechanisms underlying TAM heterogeneity and subtype specification remain unclear [11]. Elucidating these regulatory pathways is critical for reprogramming TAMs toward a tumor-inhibitory state and improving the efficacy of immunotherapeutic interventions.

*RNF135* (Ring Finger Protein 135), also known as *RIPLET*, is an E3 ubiquitin ligase originally identified for its role in antiviral responses and interferon signaling [12]. Mechanistically, *RNF135* promotes the antiviral signaling activity of the viral RNA receptor RIG-I through dual ubiquitin-dependent and -independent mechanisms. It catalyzes K63-linked ubiquitination of RIG-I to stabilize 2CARD tetramers and cross-bridges RIG-I filaments on long dsRNA, thereby amplifying the RIG-I–MAVS–TBK1/IRF3 cascade and type I/III interferon production. Beyond innate immunity, aberrant expression or silencing of *RNF135* has been reported in several solid tumors. In hepatocellular carcinoma (HCC), promoter hypermethylation results in *RNF135* deficiency, which disrupts K48-linked ubiquitination of fatty acid synthase (FASN), enhances fatty acid synthesis, and drives the accumulation of free fatty acids that activate STAT3 signaling and induce terminal CD8 T-cell exhaustion, conferring resistance to anti–PD-1 therapy. Pharmacologic inhibition of FASN can overcome this immune resistance. These findings highlight *RNF135* as a regulator at the crossroads of innate immune sensing and tumor immunometabolism [13,14,15,16]. However, its expression pattern, functional role, and regulatory impact on TAM dynamics within the GBM microenvironment have yet to be systematically investigated.

The goal of this study was to describe the expression profile and prognostic significance of *RNF135* in GBM, explain its role in creating TAM heterogeneity and immune suppression, and investigate its role in signaling networks between cells. Our results may provide new information about how GBM avoids the immune system and point to a possible target for immunotherapy that focuses on TAM.

## 2. Methods

### 2.1. Data Acquisition and Preprocessing

Transcriptomic and clinical data of GBM patients were obtained from The Cancer Genome Atlas (TCGA, https://portal.gdc.cancer.gov/, accessed on 4 October 2024) and the Chinese Glioma Genome Atlas (CGGA, http://www.cgga.org.cn/, accessed on 4 October 2024) [17,18]. Raw count and TPM expression matrices were downloaded and normalized using the “DESeq2” package in R version 1.48.2 [19]. Samples with incomplete survival information were excluded. *RNF135* expression differences between GBM and normal tissues (Figure 1B) were evaluated using the GEPIA web server (http://gepia.cancer-pku.cn/, accessed on 16 October 2024). To investigate the cell-type-specific expression pattern of *RNF135*, single-cell RNA sequencing data from newly diagnosed GBM samples were obtained from the Brain Immune Atlas (https://www.brainimmuneatlas.org/, accessed on 4 October 2024). These data were processed using the “Seurat” package, version 5.3.0 [20], and major cell populations—including tumor cells, TAMs, T cells, and oligodendrocytes—were identified using canonical marker genes. Regionally annotated bulk RNA-seq data were obtained from the Ivy Glioblastoma Atlas Project (http://glioblastoma.alleninstitute.org/, accessed on 2 October 2024) and used to assess the spatial distribution of RNF135 expression across distinct anatomical regions of GBM tissue. In accordance with the 2021 WHO classification of central nervous system tumors, only adult IDH-wildtype glioblastoma cases were included, and all IDH-mutant gliomas were excluded. Summary counts of patients and cells across all datasets are provided in Appendix A.

### 2.2. Differential Expression and Survival Analysis of RNF135

To investigate the prognostic and transcriptional impact of *RNF135* in glioblastoma, patients from the TCGA cohort were divided into *RNF135*-high and *RNF135*-low expression groups based on the median expression level. Kaplan–Meier survival analysis and univariate Cox regression were performed using the “survival” and “survminer” packages to evaluate overall survival differences between the two groups. For differential expression analysis, normalized RNA-seq expression data were analyzed using the “(version 3.64.3)” package [21]. Differentially expressed genes (DEGs) between the *RNF135*-high and *RNF135*-low groups were identified using empirical Bayes moderated t-statistics. Genes with an adjusted *p*-value < 0.05 and |log2 fold change| > 1 were considered significant. To visualize the transcriptional differences, a volcano plot was generated using the “ggplot2” (version 4.0.0) and “ggrepel” (version 0.9.6) packages, significantly upregulated and downregulated genes were color-coded, and top differentially expressed genes were labeled.

### 2.3. Functional Enrichment and GSEA

DEGs identified by the “limma” package were subjected to downstream analysis. Genes with adjusted *p*-values < 0.05 and |log2FC| > 1 were used for Gene Ontology (GO) [22] and Kyoto Encyclopedia of Genes and Genomes (KEGG) [23] enrichment analyses using the “clusterProfiler (version 4.16.0)” and “org.Hs.eg.db (version 3.21.0)” packages. Gene Set Enrichment Analysis (GSEA) [24] was performed on the ranked gene list to evaluate the enrichment of immune-related Hallmark pathways, using the “fgsea (version 1.34.2)” and “enrichplot (version 1.28.4)” packages.

### 2.4. Immune Infiltration and Correlation Analysis

To assess immune infiltration related to *RNF135* expression, we applied the MCP-counter (version 1.2.0) [25] and xCell (version 1.1.0) [26] algorithms on normalized RNA-seq data using the corresponding R packages. Based on the median *RNF135* expression, samples were stratified into high- and low-expression groups, and heatmaps were generated using the “pheatmap (version 1.0.13)” package to visualize cell type abundance patterns across groups. To further evaluate immune cell activity, we performed single-sample Gene Set Enrichment Analysis (ssGSEA) using the “GSVA (version 2.2.0)” package (Appendix A) [24]. Custom immune-related gene sets were curated and grouped by cell type. Enrichment scores for each immune cell type were calculated across samples using the “ssgsea” method with Gaussian kernel distribution. Pearson correlation coefficients were computed between *RNF135* expression and ssGSEA enrichment scores, and corresponding *p*-values were obtained via cor.test(). Significant associations (adjusted *p* < 0.05) were visualized using lollipop plots. Correlation matrices derived from MCP-counter, xCell, and ssGSEA results were visualized using the “corrplot (version 0.95)” and “ggcorrplot (version 0.1.4.1)” packages.

### 2.5. Single-Cell RNA-Seq Analysis and Cell-Type Annotation

Single-cell RNA-seq data were analyzed using the “Seurat” package (v4.0). A Seurat object was created with min.cells = 3 and min.features = 200. The dataset was log-normalized using the LogNormalize method (scale.factor = 10,000), and 2000 highly variable genes were identified with the “vst” method. After data scaling, dimensionality reduction was performed by principal component analysis (PCA) on the HVGs. Based on the elbow plot, the first 7 principal components were used for downstream analysis. Graph-based clustering was performed at a resolution of 0.4, and UMAP was applied for two-dimensional visualization. To annotate clusters, a multi-pronged strategy was employed. First, canonical marker genes were used to assign broad cell identities, and differentially expressed genes were identified for each cluster using the Wilcoxon rank-sum test (FindAllMarkers) with thresholds of log2 fold change > 0.25 and expression detected in >25% of cells. The top markers were visualized with heatmaps to facilitate cluster-level interpretation. Second, module scoring was performed using AddModuleScore based on literature-defined gene sets representing canonical GBM cell states (NPC-like, OPC-like, AC-like, MES-like) [27], and average scores were compared across clusters to aid in annotation. Third, functional enrichment analysis was conducted for each cluster using GO and KEGG pathways via the “clusterProfiler (version 4.16.0)” and “org.Hs.eg.db (version 3.21.0)” packages, helping to assign cell identities based on biological functions. Also, key signature genes uniquely enriched in individual clusters were also considered to refine subtype labels.

Based on this integrated strategy, we annotated major cell types including malignant cells, macrophages, T cells, and oligodendrocytes. Malignant cells were further stratified into six GBM transcriptional states—NPC-like, OPC-like, AC-like, MES1, MES2, and Proliferative Tumor—according to their module scores and functional status. Macrophages were subdivided into eight functionally distinct TAM subtypes, including Proliferation-TAM, IFN-TAM, Regulatory-TAM, Phago/AP-TAM, Hypoxia-TAM, Chemo-TAM, Ribo-TAM, and Transitory-TAM, guided by marker expression and enrichment-based functional annotations [28].

### 2.6. Cell–Cell Communication Analysis

Intercellular communication networks were inferred using the “CellChat” R package based on single-cell transcriptomic data [29]. Separate CellChat objects were constructed for *RNF135*-high and *RNF135*-low TAM populations. The standard pipeline was applied (subsetData—identifyOverExpressedGenes—identifyOverExpressedInteractions—computeCommunProb—computeCommunProbPathway—aggregateNet). Communication probability was computed using default parameters in computeCommunProb (population.size = TRUE, min.cells = 10), and pathways/interactions with adjusted *p* < 0.05 were retained for downstream analysis. The overall signaling network was estimated via aggregateNet. To enable group-wise comparison, CellChat objects were merged using the mergeCellChat function with dataset-specific annotations. Differential signaling activity was assessed by quantifying global communication metrics (total number and strength of interactions) and evaluating pathway-specific contributions using the rankNet function. Signaling role analysis was conducted to classify each cell population as sender, receiver, mediator, or influencer based on network centrality measures (netAnalysis_signalingRole and netAnalysis_signalingRole_heatmap). Shared and differential signaling pathways were visualized using circle plots, heatmaps, radar plots, and stacked bar charts. Focused analyses were performed on the CCL signaling pathway to investigate macrophage-intrinsic communication. Ligand–receptor interactions within the CCL pathway were identified using the subsetCommunication function in CellChat, and overall pathway structure was visualized through aggregated network plots (netVisual_aggregate) and interaction heatmaps (netVisual_heatmap). The contribution of individual ligand–receptor pairs to the CCL signaling axis was quantified using the netAnalysis_contribution function. To characterize cell-type-specific signaling roles, centrality-based sender–receiver–mediator–influencer analyses were conducted with netAnalysis_signalingRole_network and netAnalysis_signalingRole_heatmap. The expression patterns of selected ligands and receptors involved in the CCL pathway were evaluated using UMAP projections, dot plots, and violin plots, based on Seurat-defined cell identities.

### 2.7. Drug Sensitivity Prediction

Drug sensitivity prediction was performed using the “OncoPredict” R package [30]. Transcriptomic profiles of GBM samples were used as the test input, while gene expression and drug response (IC50) data from the Cancer Therapeutics Response Portal (CTRP) served as the training reference. Log-transformed IC50 values in the training set were converted back to their original scale prior to modeling. Batch effects were corrected using the empirical Bayes (ComBat) method, and low-variance genes were filtered to improve prediction accuracy. The calcPhenotype() function was applied to estimate the IC50 values of a panel of small-molecule compounds in each GBM sample. Based on median *RNF135* expression, samples were stratified into *RNF135*-high and *RNF135*-low groups. For each compound, predicted IC50 values were compared between groups using the Wilcoxon rank-sum test.

### 2.8. Statistical Analysis

All statistical analyses were conducted using R software (version 4.3.1). Comparisons between two groups were performed using the Wilcoxon rank-sum test for non-parametric data or Student’s *t*-test for normally distributed variables, as appropriate. Survival analyses were carried out using Kaplan–Meier estimation and univariate Cox proportional hazards regression, with significance assessed by the log-rank test. Pearson correlation analysis was applied to evaluate associations between *RNF135* expression and immune infiltration scores derived from MCP-counter, xCell, and ssGSEA algorithms. *P*-values were two-sided, and values less than 0.05 were considered statistically significant unless otherwise specified. Multiple testing correction was applied using the Benjamini–Hochberg method where applicable.

## 3. Results

### 3.1. RNF135 Is Upregulated in GBM and Predicts Poor Prognosis

We first screened immune-related prognostic genes across TCGA and CGGA datasets using univariate Cox regression (*p* < 0.05) and identified *RNF135* as a shared prognostic marker across three transcriptomic cohorts (Figure 1A). *RNF135* expression was significantly elevated in GBM tissues compared to normal brain tissues (Figure 1B). Kaplan–Meier survival analysis in both TCGA and CGGA cohorts showed that patients with high *RNF135* expression exhibited significantly worse overall survival (TCGA: *p* = 0.0094; CGGA: *p* = 0.0088) (Figure 1C,D).

To explore its transcriptional impact, we compared *RNF135*-high and *RNF135*-low GBM samples and identified a distinct gene signature, with numerous immune- and inflammation-related genes significantly upregulated in the *RNF135*-high group, including *DNMT1*, *MMP9*, *GSDMA*, *ITGB7*, *PLA2G7*, and *CCL2* (Figure 1E). KEGG enrichment revealed activation of immune and inflammatory pathways, such as cytokine–cytokine receptor interaction, chemokine signaling pathway, B-cell receptor signaling pathway and Th17-cell differentiation (Figure 1F). GO and GSEA analyses indicated that *RNF135*-high GBMs were enriched in innate immune activation, lysosome- and phagocytosis-related processes, and multiple inflammatory pathways including interferon response, IL6–JAK–STAT3, and TNF-α signaling (Figure 1G,H).

### 3.2. RNF135 Correlates with Immune Cell Infiltration and Is Predominantly Expressed in TAMs

Immune infiltration estimation using MCP-counter, xCell and ssGSEA algorithms showed strong positive correlations between *RNF135* expression and the abundance of macrophages, CD8+ T cells, and other immune subsets (Figure 2A–C). To further illustrate the associations between *RNF135* expression and specific immune and stromal cell populations, correlation analyses based on MCP-counter and xCell scores are provided in Appendix A. Unsupervised UMAP embedding of single-cell transcriptomic data identified four major cell populations within the glioblastoma microenvironment: macrophages, tumor cells, T cells, and oligodendrocytes (Figure 2D). Marker gene expression patterns were visualized to illustrate the transcriptional identity of each cluster (Figure 2E). Macrophages exhibited high levels of AIF1, FCER1G, and TYROBP; oligodendrocytes were marked by TF, MOG and CLDN11; T cells expressed CD3D, CD3E, and CD2 and tumor cells expressed EGFR, SOX4, and BCAN. Additional marker genes supporting cell identity assignment are shown in Appendix A. Feature plotting of *RNF135* expression across the UMAP revealed a strong and specific enrichment within the macrophage cluster (Figure 2F).

### 3.3. RNF135 Is Predominantly Expressed in Proliferation and Phagocytic TAM Subsets

To further investigate the functional context of *RNF135* at the single-cell level, we first performed unsupervised clustering of malignant cells and identified eight transcriptional clusters (Appendix A). GO and KEGG enrichment analyses were then conducted for each cluster (Appendix A), revealing distinct biological programs such as cell cycle activation, hypoxia response, extracellular matrix remodeling, and interferon signaling. Based on these enrichment patterns and module scores calculated from established gene signatures (Figure 3B), tumor cells were classified into six canonical states: AC-like, MES1, MES2, NPC-like, OPC-like, and Proliferative tumors (Figure 3A,B). These subtypes showed clear spatial separation in UMAP visualization (Figure 3C). We then analyzed the heterogeneity of TAMs and identified eight distinct transcriptional states, including Proliferation-, Phago/AP-, IFN-, Transitory-, Regulatory-, Hypoxia-, Chemo-, and Ribo-TAMs (Figure 3D,E). Among these, *RNF135* expression was most abundant in Proliferation-TAMs, followed by Phago/AP-, IFN-, and Transitory-TAM subsets (Figure 3F).

To gain deeper insights into the functional states of *RNF135*-positive vs. *RNF135*-negative TAM subsets, we performed GO enrichment analyses for each subgroup. *RNF135*-positive TAMs displayed subtype-specific enrichment patterns: Proliferation-TAMs were associated with mitotic cell cycle and chromosomal organization; Phago/AP-TAMs with metabolic processes and autophagy; IFN-TAMs with ribosome biogenesis and RNA metabolism; Regulatory-TAMs with immune regulation and leukocyte activation; and Chemo-TAMs with RNA splicing and intracellular transport (Appendix A). In contrast, *RNF135*-negative TAMs showed enrichment in stress response, cytokine signaling, and immune regulatory pathways across corresponding subtypes (Appendix A).

### 3.4. RNF135-Positive TAMs Exhibit Stronger Communication with Aggressive Tumor Cell States

To evaluate how *RNF135* expression affects macrophage-mediated intercellular communication, we performed CellChat analysis comparing *RNF135*-positive and *RNF135*-negative TAMs. *RNF135*-positive TAMs exhibited a significantly higher total number of inferred interactions (4859 vs. 3741) and stronger overall communication strength (137.4 vs. 107.1) (Figure 3I). Visualization of the communication networks revealed that *RNF135*-positive TAMs engaged more frequently and intensely with other cell types—particularly with aggressive tumor cell states such as NPC-like, Proliferative, and MES1 cells—compared to their *RNF135*-negative counterparts (Figure 3G,H).

Shared signaling pathway analysis further showed that *RNF135*-positive TAMs transmitted stronger and broader signals across multiple tumor-supportive pathways, including MHC-II, CD39, ApoE, Complement, MIF, and FN1, as reflected in both total communication weight and pathway engagement frequency (Figure 3J,K). Differential pathway activity analysis revealed that signals originating from *RNF135*-positive TAMs were preferentially received by tumor cells with aggressive phenotypes—especially NPC-like, Proliferative, and MES1 subtypes—while signals from *RNF135*-negative TAMs exhibited more limited impact (Figure 3L).

### 3.5. RNF135-Positive TAMs Exhibit Enhanced CCL Signaling and Broader Communication Profiles

Pathway-level information flow analysis showed that *RNF135*-positive TAMs had higher signaling activity than *RNF135*-negative TAMs in multiple pathways, including MHC-II, ApoE, SPP1, LAIR1, CD39, and CCL (Figure 4A). Signaling contribution heatmaps indicated that *RNF135*-positive TAMs were involved in stronger and more diverse interactions with tumor cells and other TAM subsets. The CCL pathway showed consistently higher activity in the *RNF135*-positive group (Figure 4B). To identify which TAM subtypes were involved in CCL signaling, we calculated communication probabilities among TAM subsets. *RNF135*-positive TAMs showed higher CCL signaling strength, especially involving Transitory TAM and Proliferation-TAMs (Figure 4C).

### 3.6. CCL3/CCL3L3–CCR1 Axis Is a Dominant Contributor to CCL Signaling in RNF135-Positive TAMs

To further investigate the key drivers of CCL signaling, we analyzed the roles of different cell types within the signaling network. In the *RNF135*-positive group, Transitory TAMs served as dominant signal senders and influencers, while Proliferation-TAMs acted as the main receivers and mediators. In contrast, in the *RNF135*-negative group, Transitory TAMs functioned as both senders and receivers, but had limited influence on Proliferation-TAMs (Figure 5A). Ligand–receptor pair decomposition revealed that CCL3–CCR1 and CCL3L3–CCR1 were the major contributors to the overall CCL pathway activity (Figure 5B). We then examined the expression patterns of these genes across cell subtypes. CCL3, CCL3L3, and CCR1 were predominantly expressed in Transitory- and Proliferation-TAMs (Figure 5C). Violin plots confirmed these findings, showing the highest expression levels in the same TAM subsets (Figure 5D). UMAP feature plots further demonstrated spatial co-localization of CCL3, CCL3L3, and CCR1 within the TAM compartment (Figure 5E).

### 3.7. RNF135 Expression Is Enriched in Microvascular Proliferation Regions and Associated with Differential Drug Sensitivity

To explore the spatial distribution of *RNF135* expression in glioblastoma tissue, we analyzed spatial transcriptomic data from the Ivy Glioblastoma Atlas Project. Principal component analysis revealed distinct regional clustering of samples, and *RNF135* expression was highly enriched in the CT-mvp region, which corresponds to microvascular proliferation (Figure 5F).

Finally, to evaluate the therapeutic relevance of *RNF135* expression, we performed drug sensitivity prediction using transcriptomic profiles. Analysis of a panel of targeted agents revealed significant differences in predicted response between the *RNF135* high- and low-expression groups. Among the tested compounds, Selumetinib—a MEK inhibitor—exhibited the greatest difference in predicted response, with significantly higher sensitivity observed in the *RNF135* high-expression group (Figure 5G).

## 4. Discussion

In this study, we identified *RNF135* as a key immunoregulatory molecule in GBM, beyond its established tumor-intrinsic oncogenic functions. *RNF135* was significantly upregulated in TAMs, particularly within proliferation, phagocytic, and transitional subtypes. These *RNF135*-enriched TAMs exhibited enhanced intercellular communication, especially via the CCL3/CCL3L3–CCR1 axis, and were associated with poor patient prognosis. Our findings reveal a novel dimension of *RNF135* activity—namely, its role in shaping an immunologically active yet immunosuppressive microenvironment—thereby advancing our understanding of immune evasion mechanisms in GBM.

Previous studies have primarily focused on the oncogenic role of *RNF135* in glioma cells. Gu et al. reported that *RNF135* promotes tumor proliferation by facilitating the ubiquitin-mediated degradation of p21, accelerating cell cycle progression [31]. Similarly, Zhang et al. showed that *RNF135*, as a downstream target of miR-485-3p, activates the MAPK/ERK signaling pathway to drive glioma cell growth and migration [32]. Consistent with these mechanistic insights, our transcriptome-based drug sensitivity analysis revealed that *RNF135*-high GBMs were more susceptible to Selumetinib, a selective MEK1/2 inhibitor that targets the MAPK/ERK cascade [33]. This suggests that *RNF135* may serve as a biomarker for MAPK pathway–dependent gliomas and support the potential utility of MEK inhibition in *RNF135*-overexpressing tumors.

Our study further reveals that *RNF135*-positive TAMs predominantly engage in CCL signaling, especially through the CCL3/CCL3L3–CCR1 axis, which emerged as a dominant contributor to TAM–TAM and TAM–tumor cell communication. *RNF135* expression was highest in proliferation and transitory TAM subsets, which correspondingly showed elevated expression of CCL3, CCL3L3, and CCR1. This suggests that *RNF135* may transcriptionally or post-transcriptionally promote CCL ligand production, facilitating autocrine and paracrine loops that reinforce TAM-mediated immunosuppression. Importantly, CCR1 signaling has been implicated in the recruitment and retention of myeloid cells within tumors, promotion of M2-like polarization, and evasion of T cell-mediated cytotoxicity [34,35,36]. Thus, the RNF135–CCL3/CCR1 axis may represent a critical hub by which glioblastoma orchestrates an immunosuppressive and tumor-supportive microenvironment.

Beyond its classical function as an E3 ubiquitin ligase, *RNF135* (also known as *RIPLET*) has been shown to amplify antiviral innate immune responses by promoting RIG-I clustering and signaling. Cadena et al. revealed that *RNF135* catalyzes K63-linked polyubiquitination and physically cross-bridges filamentous RIG-I complexes, enhancing downstream signaling in an RNA-length–dependent manner [16]. Although RIG-I was not directly evaluated in our study, the strong communication network and subtype-specific clustering of *RNF135*-high TAMs raise the possibility that similar oligomerization-based mechanisms may facilitate the assembly of immunosuppressive signaling hubs. Alternatively, *RNF135* may promote NF-κB– or IRF-mediated transcription of chemokines such as CCL3, thereby reinforcing TAM–TAM and TAM–tumor interactions.

Mechanistically, this immunoregulatory role appears to be context dependent. For instance, in hepatocellular carcinoma, epigenetic silencing of *RNF135* induces lipid metabolic reprogramming and STAT3 activation, leading to T cell exhaustion and resistance to immune checkpoint blockade [15]. While *RNF135* is silenced in HCC but upregulated in GBM, both scenarios converge on immune dysfunction through distinct pathways—metabolic vs. chemokine-mediated suppression—underscoring the context-dependent immunoregulatory roles of *RNF135* across tumor types.

Taken together, *RNF135* may act as a dual-function effector in GBM, driving tumor proliferation via MAPK signaling and reinforcing macrophage-mediated immunosuppression through CCL3/CCR1 communication. These findings position *RNF135* as both a mechanistic biomarker and a potential therapeutic target. Given the availability of CCR1 antagonists and MEK inhibitors in clinical development, combinatorial targeting of *RNF135*-driven pathways may offer novel immunotherapeutic strategies for *RNF135*-high GBM.

This study has several limitations. First, our analyses were based on publicly available bulk and single-cell transcriptomic datasets, without in vitro or in vivo functional validation. While the consistency of our findings across independent cohorts strengthens confidence in the results, experimental studies are required to confirm the causal role of *RNF135* in modulating TAM phenotypes, intercellular signaling, and therapeutic susceptibility. Second, we did not include proteomic data to validate *RNF135* expression at the protein level within specific cell populations. Future studies incorporating immunohistochemistry, mass spectrometry, or single-cell proteomic approaches will be necessary to confirm the cellular localization and functional relevance of *RNF135*. Third, the predicted sensitivity to selumetinib was derived from transcriptome-based computational modeling rather than direct experimental testing, and therefore should be interpreted with caution until validated in preclinical models. Nonetheless, to the best of our knowledge, this is the first study to report the immunological role of *RNF135* in GBM, and we hope that our findings will provide a valuable framework for subsequent mechanistic and translational investigations.

## 5. Conclusions

In summary, our findings uncover a previously unrecognized immunosuppressive role of RNF135 in GBM TAMs and suggest that RNF135 may serve as a functional effector and predictive biomarker for both immune evasion and targeted therapy susceptibility. These insights provide a conceptual framework for future studies aiming to integrate innate immune signaling components into precision immunotherapy strategies for GBM.

## Figures and Tables

**Figure 1 cancers-17-03271-f001:**
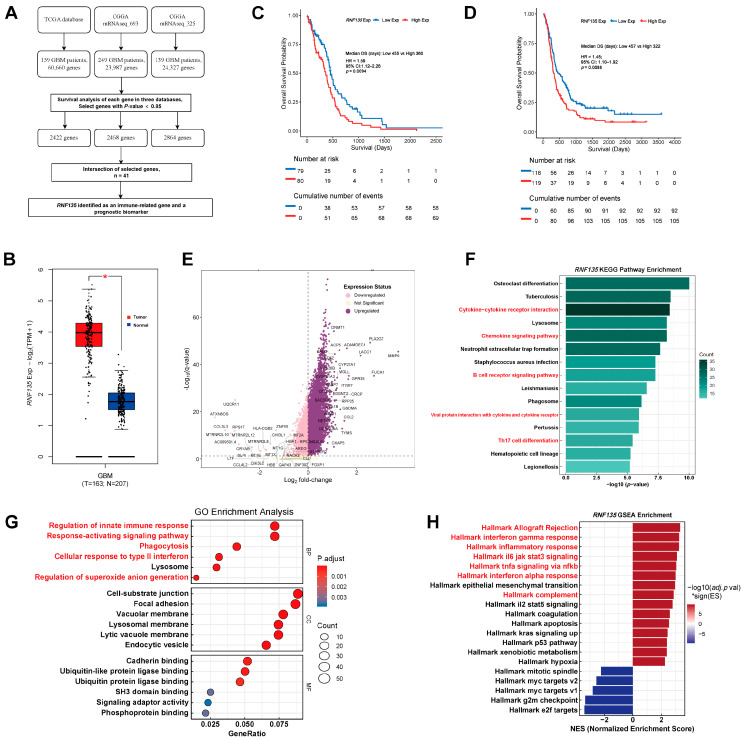
*RNF135* is upregulated in glioblastoma and predicts poor prognosis. (**A**) Workflow for identifying immune-related prognostic genes across TCGA and CGGA datasets. *RNF135* was identified as a shared prognostic biomarker. (**B**) *RNF135* expression was significantly elevated in GBM tumors compared to normal brain tissues (TCGA). (**C**,**D**) Kaplan–Meier survival analysis of TCGA (**C**) and CGGA (**D**) cohorts showing that high *RNF135* expression is associated with significantly worse overall survival. (**E**) Volcano plot of differentially expressed genes between *RNF135*-high and -low groups in GBM, immune and inflammatory related genes were upregulated in the *RNF135*-high group. (**F**) KEGG pathway enrichment analysis of *RNF135*-high GBMs shows significant enrichment in immune-related signaling pathways. (**G**) GO enrichment analysis of DEGs reveals upregulation of innate immune responses, phagocytosis, lysosome, and related processes. (**H**) GSEA demonstrates enrichment of immune related hallmark pathways (e.g., interferon gamma, IL6/JAK/STAT3, TNF-α signaling) in the *RNF135*-high group. * indicates statistical significance (*p* < 0.05).

**Figure 2 cancers-17-03271-f002:**
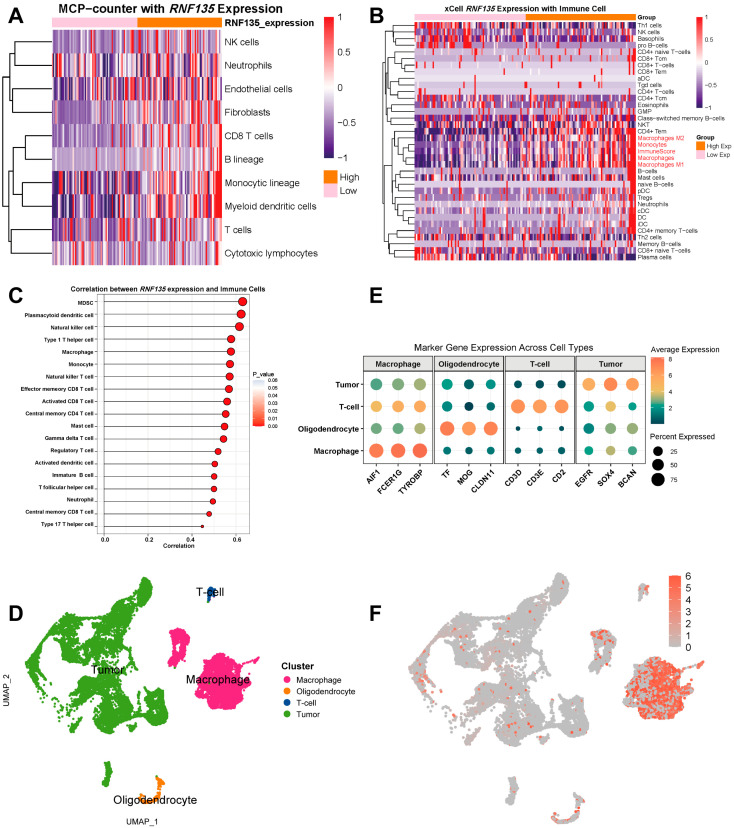
*RNF135* correlates with immune infiltration and is predominantly expressed in TAMs. (**A**) Heatmap showing correlations between *RNF135* expression and various immune cell populations estimated by MCP-counter. (**B**) xCell analysis confirms higher abundance of immune cells, including macrophages and T cells, in *RNF135*-high GBMs. (**C**) Correlation plot from ssGSEA showing strong associations between *RNF135* expression and multiple immune cell types. (**D**) UMAP visualization of major cell clusters identified in GBM single-cell RNA-seq data: macrophages, tumor cells, T cells, and oligodendrocytes. (**E**) Dot plot showing marker gene expression across cell types. (**F**) Feature plot shows *RNF135* expression is enriched specifically in the macrophage cluster.

**Figure 3 cancers-17-03271-f003:**
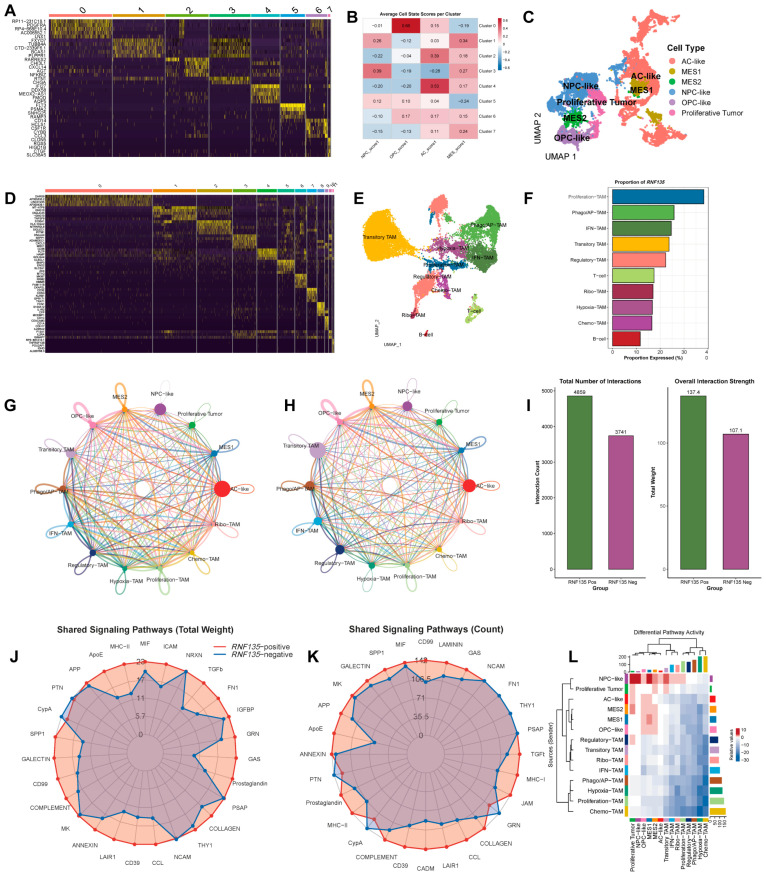
*RNF135* is enriched in proliferation and phagocytic TAM subsets and enhances TAM–tumor communication. (**A**) Heatmap of marker gene expression in eight malignant tumor clusters. (**B**) Heatmap of gene signature scores used to define canonical tumor subtypes. (**C**) UMAP visualization of tumor subtypes, including NPC-like, MES1, MES2, AC-like, OPC-like, and proliferative tumors. (**D**,**E**) Heatmap (**D**) and UMAP (**E**) of transcriptionally defined TAM subsets, including Proliferation-, Phago/AP-, IFN-, Transitory-, Regulatory-, Hypoxia-, Chemo-, and Ribo-TAMs. (**F**) Bar plot showing *RNF135* expression proportion across TAM subtypes, with highest levels in Proliferation-TAMs. (**G**,**H**) CellChat network diagrams comparing intercellular communication patterns between *RNF135*-positive (**G**) and *RNF135*-negative (**H**) TAMs. (**I**) Bar plots summarizing the total number of interactions and communication strength, which are higher in *RNF135*-positive TAMs. (**J**,**K**) Radar plots showing signaling pathways shared across TAM–tumor networks, with *RNF135*-positive TAMs engaging more actively in immunomodulatory pathways (**J**: total weight; **K**: count). (**L**) Heatmap of differentially active signaling pathways across TAM subtypes, highlighting preferential signaling from *RNF135*-positive TAMs to aggressive tumor states.

**Figure 4 cancers-17-03271-f004:**
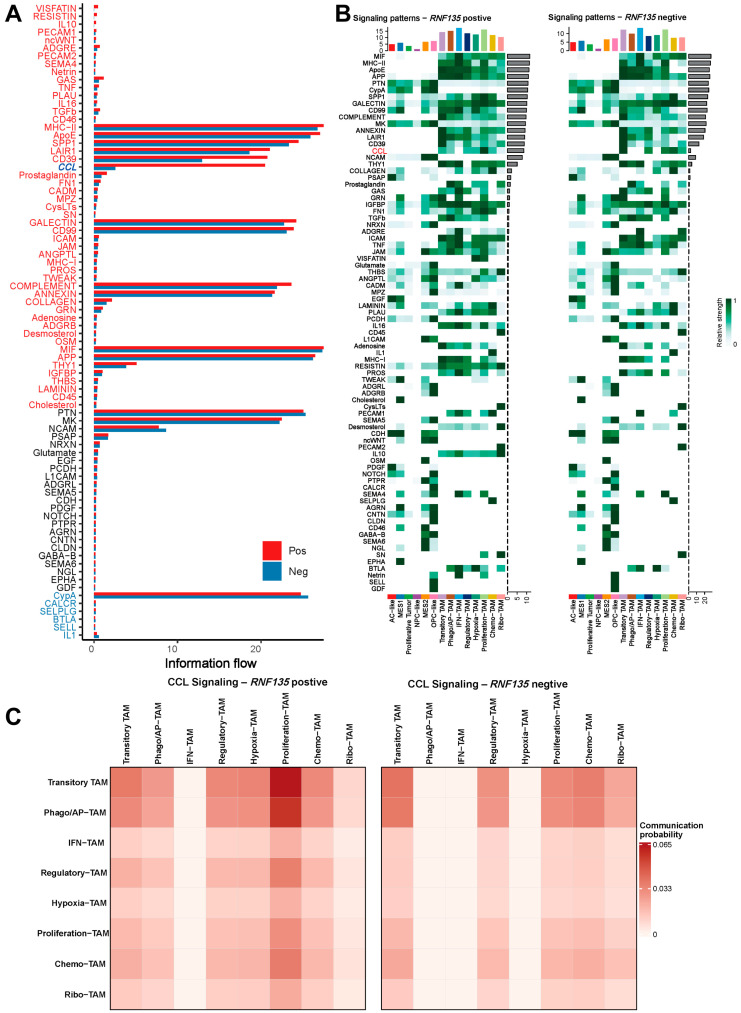
*RNF135*-positive TAMs exhibit enhanced CCL signaling and inter-TAM communication. (**A**) Information flow comparison showing increased signaling activity in *RNF135*-positive TAMs across pathways, especially CCL. (**B**) Heatmap comparing CCL-related signaling pathway strength across TAM subtypes between *RNF135*-positive and -negative groups. (**C**) Heatmaps showing pairwise CCL signaling probabilities among TAM subtypes in *RNF135*-positive vs. negative conditions.

**Figure 5 cancers-17-03271-f005:**
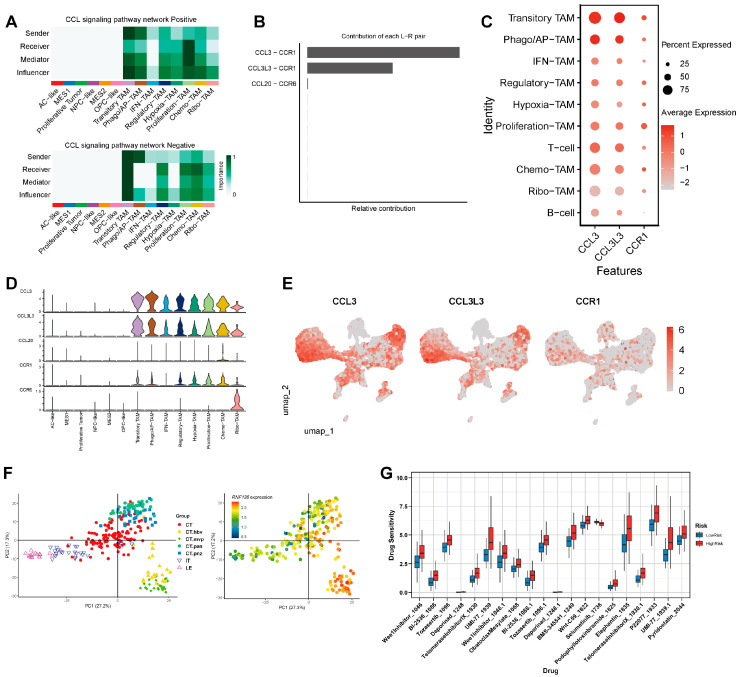
CCL3/CCL3L3–CCR1 axis mediates enhanced signaling in *RNF135*-positive TAMs. (**A**) CCL signaling role analysis identifies *RNF135*-positive Transitory-TAMs as major senders and Proliferation-TAMs as dominant receivers. (**B**) Contribution analysis of ligand–receptor pairs highlights CCL3–CCR1 and CCL3L3–CCR1 as the top contributors to CCL signaling. (**C**) Dot plot of *CCL3, CCL3L3,* and *CCR1* expression across TAM and other immune cell subtypes. (**D**) Violin plots showing expression distribution of *CCL3*, *CCL3L3*, *CCR1*, and *CCR6* across TAM subsets. (**E**) UMAP feature plots displaying spatial co-localization of *CCL3, CCL3L3,* and *CCR1* in TAM clusters. (**F**) PCA of Ivy GAP spatial transcriptomic data showing high *RNF135* expression in the microvascular proliferation (CT-mvp) region. (**G**) Boxplots comparing predicted drug sensitivity between *RNF135*-high and -low GBMs, with Selumetinib showing higher sensitivity in the *RNF135*-high group.

## Data Availability

The data supporting the findings of this study are available upon reasonable request from the corresponding author. All custom R code for multi-omics integration, scoring, model construction, and statistical analysis has been deposited in GitHub repository (https://github.com/cielowq/RNF135-Marks-Chemokine-Ligand-Enriched-Macrophage-Tumor-Interactions-in-the-Glioblastoma-TME.git) (accessed on 7 October 2025), and Harvard Dataverse repository (https://doi.org/10.7910/DVN/EZKS7D) (accessed on 7 October 2025).

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
