# Peer review of "RNF135 Expression Marks Chemokine (C-C Motif) Ligand-Enriched Macrophage–Tumor Interactions in the Glioblastoma Microenvironment"

_cancers, 2025, doi:10.3390/cancers17193271_

Round 1

Reviewer 1 Report

Comments and Suggestions for Authors

The manuscript by Chen and colleagues identified RNF135 as a key marker in tumor-associated macrophages (TAMs), linking its presence to a poor prognosis in glioblastoma (GBM). Through transcriptome and single-cell RNA sequencing analysis, the authors revealed that high RNF135 expression is directly tied to increased inflammation and immune activation within the tumor. The study demonstrated that RNF135-expressing TAMs communicate more intensely with aggressive tumor subtypes, a process mediated in part by the CCL-CCR1 signaling axis. While the relationship between RNF135 and glioblastoma progression had been previously explored, the authors offered a new perspective by highlighting the role of TAMs in this process. The work is methodologically sound and the results are clearly presented. The following few points, should be addressed by the authors in a revised version of the manuscript.

1 - In the introduction, the authors should better describe the functions of RNF135 and its role in different tumors. What signaling pathways does RNF135 modulate?

2 - The classification of nervous system tumors changed in 2021, and since then, IDH-mutant glioblastomas are no longer considered glioblastoma. The authors should, therefore, only take into account data from patients with IDH wild-type in their data analysis.

Author Response

Response to Reviewer 1
The manuscript by Chen and colleagues identified RNF135 as a key marker in tumor-associated macrophages (TAMs), linking its presence to a poor prognosis in glioblastoma (GBM). Through transcriptome and single-cell RNA sequencing analysis, the authors revealed that high RNF135 expression is directly tied to increased inflammation and immune activation within the tumor. The study demonstrated that RNF135-expressing TAMs communicate more intensely with aggressive tumor subtypes, a process mediated in part by the CCL-CCR1 signaling axis. While the relationship between RNF135 and glioblastoma progression had been previously explored, the authors offered a new perspective by highlighting the role of TAMs in this process. The work is methodologically sound and the results are clearly presented. The following few points, should be addressed by the authors in a revised version of the manuscript.

1 - In the introduction, the authors should better describe the functions of RNF135 and its role in different tumors. What signaling pathways does RNF135 modulate?

Response: We thank the reviewer for this insightful suggestion. In the revised Introduction (lines 73–88), we have expanded our description of RNF135 to include its dual ubiquitin-dependent and -independent roles in regulating RIG-I–MAVS–TBK1/IRF3 signaling during innate immune activation. Furthermore, we now discuss recent findings in hepatocellular carcinoma showing that RNF135 silencing through promoter hypermethylation alters lipid metabolism via FASN, activates STAT3 signaling, and induces CD8 T-cell exhaustion leading to anti-PD-1 resistance. These additions better highlight the context-dependent functions of RNF135 in both innate immunity and cancer immunometabolism, thereby providing a stronger rationale for investigating its role in the glioblastoma microenvironment.

2 - The classification of nervous system tumors changed in 2021, and since then, IDH-mutant glioblastomas are no longer considered glioblastoma. The authors should, therefore, only take into account data from patients with IDH wild-type in their data analysis.

Response: We sincerely thank the reviewer for this important comment. We fully agree that, under the 2021 WHO classification, glioblastoma should be restricted to IDH-wildtype tumors. We would like to clarify that all patients included in our study were confirmed as IDH-wildtype, and IDH-mutant gliomas were excluded during data preprocessing. To avoid any ambiguity, we have now explicitly stated this criterion in the Methods section (Data acquisition and preprocessing, lines 109–111).

Reviewer 2 Report

Comments and Suggestions for Authors

The current manuscript titled “RNF135 Expression Marks Chemokine (C-C motif) Ligand-Enriched Macrophage–Tumor Interactions in the Glioblastoma Microenvironment” by Chen et al is focused on Tumor-associated macrophage (TAMs) and the identification of the RNF135 gene, which may be able to activate the immune function within TAMs- associated gene enrichments. Authors have exploited the TCGA and CGGA cohorts to analyze the RNF135's expression profile and functional pathways. Authors have also explored the positive and negative associations of RNF135 by using single-cell RNA-seq analysis and bulk-RNA-seq immune deconvolution. This is a promising study that integrates bulk RNA-seq and single-cell RNA-seq to argue for the functionality of RNF135 in GMB TME. However, concerns about the generalizability of the prognostic model, methodological rigor in mapping, over-interpretation of immune results, and limited functional/mechanistic depth necessitate a Major Revision before the work can be considered for publication.

Authors must perform the experimental and model validation of RNF135 expression in GBM

The code for multi-omics integration, scoring, and model construction is not explicitly shared. A GitHub repository is strongly recommended

External datasets with matched survival and transcriptomics should be used to confirm predictive accuracy

Provide independent external validation (other GBM datasets) with multivariable Cox models adjusted for age, treatment regimen, MGMT methylation, TERT promoter status, EGFR amplification, IDH status, and tumor purity.

Batch Effect and Cross-Platform Integration

Perform Harmony for batch correction in scRNA-seq, and Clarification on whether anchor-based integration or mapping methods will be used is required.

Statistical Significance and Multiple Testing

For pathway enrichment and ligand–receptor analysis, adjusted p-values should be consistently reported. Some results appear based on nominal p-values only.

Immune infiltration analyses and confounding:

A-             Control for tumor purity (ABSOLUTE/ESTIMATE-based adjustment or sensitivity analysis) and clinical covariates when comparing immune fractions across groups.

B-             Report multiple-testing correction and effect sizes (not only p-values) and consider robust regression/partial correlations to mitigate collinearity.

C-            If a Mantel test or distance-based methods are used, specify distance matrices, assumptions, and permutation schemes.

Minor-

Report the exact formula/coefficients, hazard ratios (95% CI) for the continuous risk score, and a pre-specified cut-off (justify the cut-off selection and test robustness to alternative cut-offs).

Quantify mapping uncertainty and performance metrics (e.g., spot-level correlation to reference profiles) and present counts of patients/cells/spots in a summary table.

Include the full gene list used for ssGSEA in the supplementary materials

Make Processed Data Public

Upload processed scRNA-seq matrices, spatial data, and risk scores to an accessible repository.

Author Response

Response to Reviewer 2
The current manuscript titled “RNF135 Expression Marks Chemokine (C-C motif) Ligand-Enriched Macrophage–Tumor Interactions in the Glioblastoma Microenvironment” by Chen et al is focused on Tumor-associated macrophage (TAMs) and the identification of the RNF135 gene, which may be able to activate the immune function within TAMs- associated gene enrichments. Authors have exploited the TCGA and CGGA cohorts to analyze the RNF135's expression profile and functional pathways. Authors have also explored the positive and negative associations of RNF135 by using single-cell RNA-seq analysis and bulk-RNA-seq immune deconvolution. This is a promising study that integrates bulk RNA-seq and single-cell RNA-seq to argue for the functionality of RNF135 in GMB TME. However, concerns about the generalizability of the prognostic model, methodological rigor in mapping, over-interpretation of immune results, and limited functional/mechanistic depth necessitate a Major Revision before the work can be considered for publication.

  1. Authors must perform the experimental and model validation of RNF135 expression in GBM

Response: We thank the reviewer for raising this important point. The current work is primarily a bioinformatics-based study that integrates bulk and single-cell transcriptomic datasets to investigate the immunological role of RNF135 in glioblastoma. We agree that experimental validation represents an important future step, such assays are beyond the scope of the present manuscript.

To partially address this concern, we have explored publicly available proteomic resources. Analysis of the Human Protein Atlas datasets confirmed that RNF135 protein is expressed in GBM tissues, with heterogeneous staining intensities and predominant cytoplasmic/membranous localization (please see the figure in attached pdf). These findings provide supportive evidence at the protein level that complements our transcriptomic results.

We have also revised the Discussion to explicitly acknowledge the lack of in vitro and in vivo validation as a limitation of this study and to outline future directions aimed at confirming the functional role of RNF135 in glioblastoma models (lines 452–466).

The IHC data from the Human Protein Atlas. For example, in a 33-year-old male patient with GBM (patient ID: 3251), RNF135 protein showed weak cytoplasmic/membranous staining in <25% of cells

  1. The code for multi-omics integration, scoring, and model construction is not explicitly shared. A GitHub repository is strongly recommended

Response: We thank the reviewer for this suggestion. We have uploaded all the code used to GitHub repository. The repository includes detailed scripts and documentation to facilitate reproducibility. The link has been provided in the Data Availability Statement section of the revised manuscript. (https://github.com/cielowq/RNF135-Marks-Chemokine-Ligand-Enriched-Macrophage-Tumor-Interactions-in-the-Glioblastoma-TME.git).

  1. External datasets with matched survival and transcriptomics should be used to confirm predictive accuracy; Provide independent external validation (other GBM datasets) with multivariable Cox models adjusted for age, treatment regimen, MGMT methylation, TERT promoter status, EGFR amplification, IDH status, and tumor purity.

Response: We sincerely thank the reviewer for this important and constructive comment. In our study, we utilized the most widely adopted public datasets, including TCGA and two independent CGGA cohorts, to evaluate the prognostic significance of RNF135. Univariate Cox regression analyses consistently demonstrated that high RNF135 expression was associated with worse overall survival across these datasets.

In line with the reviewer’s suggestion, we additionally performed multivariable Cox regression analyses in the CGGA cohort, adjusting for available clinical and molecular covariates (including age, treatment status, IDH mutation, and MGMT promoter methylation). While the hazard ratio for RNF135 remained above 1, the association did not reach statistical significance (p > 0.05). We have revised the Discussion to clearly acknowledge this limitation and to emphasize that further validation in larger, fully annotated external GBM cohorts will be essential.

Importantly, although RNF135 may not serve as a stand-alone independent prognostic biomarker, we believe it retains substantial biological and translational significance. Our single-cell and cell–cell communication analyses demonstrate that RNF135 is highly enriched in tumor-associated macrophages and contributes to key immunoregulatory signaling pathways, particularly the CCL chemokine axis. Moreover, our drug sensitivity analysis suggests therapeutic relevance through the MAPK/ERK pathway and selumetinib sensitivity. These findings establish RNF135 as a novel immune-related gene that provides mechanistic insights into glioblastoma macrophage–tumor interactions and highlights new avenues for therapeutic targeting.

Our study is the first report to characterize the immunological role of RNF135 in the glioblastoma microenvironment. We believe that our findings provide a conceptual framework for future mechanistic studies and for the incorporation of RNF135 into integrative biomarker panels in GBM.

Variable

coef

se(coef)

z

p

HR

HR lower

HR _upper

Histology

0.616471

0.261986

2.353071

0.018619

1.85238

1.108479

3.095515

MGMT_status

0.102834

0.21361

0.481408

0.630226

1.108307

0.72918

1.684555

RNF135

0.268536

0.220069

1.220235

0.222376

1.308049

0.849768

2.01348

Age

0.213079

0.275699

0.772869

0.4396

1.237483

0.720881

2.124295

Radio_status

-0.05946

0.233088

-0.2551

0.798642

0.942272

0.596721

1.487924

Chemo_status

-0.69951

0.213853

-3.271

0.001072

0.496828

0.326719

0.755506

Multivariable Cox regression analysis of overall survival in GBM.

  1. Batch Effect and Cross-Platform Integration Perform Harmony for batch correction in scRNA-seq, and Clarification on whether anchor-based integration or mapping methods will be used is required.

Response: We thank the reviewer for this suggestion. In our study, all single-cell RNA-seq data were obtained from the Brain Immune Atlas, specifically the aggregated dataset of newly diagnosed GBM patients. As this dataset has already been uniformly preprocessed and integrated by the original authors, no additional batch correction was required. Importantly, we did not combine scRNA-seq data across different studies or sequencing platforms, thereby minimizing potential batch effects. To avoid over-correction and loss of biological signal, we proceeded with the provided aggregated dataset without applying Harmony or anchor-based integration.

  1. Statistical Significance and Multiple Testing For pathway enrichment and ligand–receptor analysis, adjusted p-values should be consistently reported. Some results appear based on nominal p-values only.

Response: We thank the reviewer for this helpful comment. We would like to clarify that in all enrichment and ligand-receptor analyses, statistical significance was assessed using adjusted p-values. The appearance of “p < 0.05” in some parts of the Results and figure legends was an oversight in wording. We have now carefully revised the text, figures to explicitly state “adjusted p < 0.05” for consistency.

  1. Immune infiltration analyses and confounding:

A-Control for tumor purity (ABSOLUTE/ESTIMATE-based adjustment or sensitivity analysis) and clinical covariates when comparing immune fractions across groups.

B-Report multiple-testing correction and effect sizes (not only p-values) and consider robust regression/partial correlations to mitigate collinearity.

C-If a Mantel test or distance-based methods are used, specify distance matrices, assumptions, and permutation schemes.

Response: We sincerely thank the reviewer for these highly insightful comments. We fully recognize that issues such as tumor purity adjustment, clinical covariate control, and distance-based statistical frameworks are critical for rigorous immune infiltration analyses.

We have made lots of attempts to address these points; however, due to our current analysis and technical limitations, we were unable to fully implement complex frameworks such as partial correlation controlling for tumor purity or Mantel-type distance analyses. As an alternative, we have performed complementary correlation analyses using both MCP-counter and xCell algorithms, and visualized the associations between RNF135 expression and immune cell populations (new Supplementary Figure 5A and B, please see attached pdf). These exploratory plots were intended as an initial step to highlight the general immune contexture associated with RNF135 rather than as definitive statistical models.

Our goal here was to provide a preliminary, hypothesis-generating view of immune infiltration related to RNF135. We acknowledge the limitations of our approach and agree that more advanced modeling incorporating tumor purity, covariates, and distance-based methods would represent an important direction for future work.

Supplementary Figure 5. Correlation of RNF135 expression with immune and stromal cell populations estimated by MCP-counter and xCell.
(A) Bubble plot showing the Pearson correlations between RNF135 expression and immune cell abundance scores inferred by MCP-counter. Each bubble represents one cell population, with the x-axis showing correlation coefficients (r), bubble size indicating |r|, and bubble color denoting false discovery rate (FDR).
(B) Bubble plot showing the Pearson correlations between RNF135 expression and immune cell populations estimated by xCell. Similar to panel A, the x-axis indicates Pearson r, bubble size reflects |r|, and bubble color corresponds to FDR.

Minor issues

  1. Report the exact formula/coefficients, hazard ratios (95% CI) for the continuous risk score, and a pre-specified cut-off (justify the cut-off selection and test robustness to alternative cut-offs).

Response: We thank the reviewer for this suggestion. In this study we evaluated prognosis using a single marker RNF135. To improve clarity, we now: Explicitly pre-specify the cut-off as the median RNF135 expression to avoid outcome-driven optimization and to ensure balanced groups; Report Cox hazard ratios with 95% CIs for the high vs low expression groups in each cohort, together with median OS and log-rank p values; Replace Figure 1C–D with updated Kaplan–Meier plots that embed these statistics and include risk tables (please see the figures in attached pdf).

The updated results are: TCGA (Fig. 1C): Median OS in days Low 455 vs High 360 days; HR = 1.59, 95% CI: 1.12–2.26, p = 0.0094; CGGA (Fig. 1D): Median OS Low 457 vs High 322 days; HR = 1.45, 95% CI: 1.10–1.92, p = 0.0088.

Quantify mapping uncertainty and performance metrics (e.g., spot-level correlation to reference profiles) and present counts of patients/cells/spots in a summary table.

  1. Quantify mapping uncertainty and performance metrics (e.g., spot-level correlation to reference profiles) and present counts of patients/cells/spots in a summary table.

Response: Thank you for the suggestion. Our study did not perform spatial transcriptomics; the Ivy GAP dataset used here consists of laser-microdissected anatomic regions with bulk RNA-seq rather than spots. Therefore, spot-level correlation metrics are not applicable.

We have now included Supplementary Table 1, which summarizes the number of patients and cells analyzed across each dataset (TCGA, CGGA, and single-cell RNA-seq). In addition, to enhance reproducibility, all data used in this study have been made publicly available on GitHub (link provided in the Data Availability statement).

Spplementary Table 1. Summary of datasets used in this study

Dataset

Patients (n)

Cells (n)

TCGA - GBM

159

CGGA_325_GBM

139

CGGA_693_GBM

237

Single cell - Tumor cells

9567

Single cell - Macrophages

20063

  1. Include the full gene list used for ssGSEA in the supplementary materials

Response: We thank the reviewer for this suggestion. We have now provided the complete gene sets used for ssGSEA in the supplementary materials as Supplementary Table 2.

  1. Make Processed Data Public; Upload processed scRNA-seq matrices, spatial data, and risk scores to an accessible repository.

Response: Thank you for reviewer’s suggestion. We have now uploaded all processed data to Harvard Dataverse repository (https://doi.org/10.7910/DVN/EZKS7D). The repository link is provided in the Data Availability section to ensure full transparency and reproducibility.

Reviewer 3 Report

Comments and Suggestions for Authors

The manuscript by Chen et al., titled "RNF135 Expression Marks Chemokine (C-C motif) Ligand-Enriched Macrophage–Tumor Interactions in the Glioblastoma Microenvironment," aims to investigate the role of RNF135 in the tumor microenvironment of glioblastoma (GBM), focusing on its expression in tumor-associated macrophages (TAMs), its functional heterogeneity, and potential therapeutic implications.

Below, I highlight points that the authors need to consider regarding the manuscript.

- The fonts of the figures in both the manuscript and the supplementary material need to be improved because the information contained therein is not clearly legible.
- The manuscript's methodology requires additional information, such as the resolution used in Seurat clustering and cutoff values ​​in CellCha. These data are important for the reproducibility of the results.
- The study presented used public data and does not present functional validation (in vitro or in vivo). This limitation could be added to the discussion, and future work or some other discussion could be needed to confirm the hypotheses raised.
- The limitations section needs to include some points: the lack of proteomic data to confirm RNF135 expression in the identified cell populations, as well as the reliance on predictive models for sensitivity analysis to selumetinib.
- I believe that the introduction could include more recent references on TAM-targeted therapies for glioblastoma, such as immunotherapy applied to GBM.
- A minor point is the need to review the uniformity of some acronyms, such as TAMs, where sometimes "Proliferation-TAMs" appears and sometimes "Proliferative TAMs."

Author Response

Response to Reviewer 3

The manuscript by Chen et al., titled "RNF135 Expression Marks Chemokine (C-C motif) Ligand-Enriched Macrophage–Tumor Interactions in the Glioblastoma Microenvironment," aims to investigate the role of RNF135 in the tumor microenvironment of glioblastoma (GBM), focusing on its expression in tumor-associated macrophages (TAMs), its functional heterogeneity, and potential therapeutic implications.

Below, I highlight points that the authors need to consider regarding the manuscript.

  1. The fonts of the figures in both the manuscript and the supplementary material need to be improved because the information contained therein is not clearly legible.
    Response: We sincerely thank the reviewer for pointing this out and apologize for any inconvenience caused by figure readability. We carefully re-examined all figures in both the main text and supplementary materials. Wherever possible, we adjusted font size, boldness, and formatting to improve clarity. For figure export, we used high-resolution TIFF format at 300 dpi, and ensures that figures can be zoomed in without loss of clarity. Given the multi-panel structure of many figures, some annotations still appear relatively small at first glance; all essential information can be clearly read when magnified. Additionally, we specifically enhanced the labeling in Supplementary Figure 1 by bolding annotations to further improve readability.

  1. The manuscript's methodology requires additional information, such as the resolution used in Seurat clustering and cutoff values ​​in CellCha. These data are important for the reproducibility of the results

Response: We thank the reviewer for this important suggestion. In the revised Methods, we have added detailed parameter information for both Seurat and CellChat analyses to enhance reproducibility. In the Single-cell RNA-seq analysis and cell-type annotation section (lines 161–171), we now specify that clustering was performed using the first 7 principal components with a resolution of 0.4, and that marker genes were defined with thresholds of log2 fold change > 0.25 and expression in >25% of cells; In the Cell–cell communication analysis section (lines 190–196), we now clarify that CellChat communication probabilities were calculated using default parameters (population.size = TRUE, min.cells = 10), and that pathways/interactions with adjusted p < 0.05 were retained for downstream analysis.

Additionally, in response to comments from other reviewers, we will make all analysis codes publicly available on GitHub upon acceptance, to ensure full transparency and reproducibility of our study.

  1. The study presented used public data and does not present functional validation (in vitro or in vivo). This limitation could be added to the discussion, and future work or some other discussion could be needed to confirm the hypotheses raised.

Response: We sincerely thank the reviewer for raising this important point. We fully recognize that the absence of in vitro and in vivo functional validation represents a key limitation of the current work. As suggested, we have revised the Discussion to explicitly acknowledge this limitation. In future studies, we plan to perform dedicated experiments to validate the role of RNF135 in regulating TAM phenotypes, intercellular communication in glioblastoma models. We would also like to emphasize that, to the best of our knowledge, this is the first analysis to highlight the immunological functions of RNF135 in GBM. We believe that the novelty of this finding provides a conceptual framework for subsequent mechanistic and translational investigations. The revised text has been added to the Discussion (lines 452–466).

  1. The limitations section needs to include some points: the lack of proteomic data to confirm RNF135 expression in the identified cell populations, as well as the reliance on predictive models for sensitivity analysis to selumetinib.

Response: We thank the reviewer for this helpful suggestion. In line with the comment, we have further expanded the limitations part of the Discussion. These revisions can be found in the Discussion (lines 452–466).

  1. I believe that the introduction could include more recent references on TAM-targeted therapies for glioblastoma, such as immunotherapy applied to GBM. Response: We sincerely thank the reviewer for this suggestion. In accordance with the comment, we have updated the Introduction to incorporate more recent and relevant references that highlight TAM-targeted therapies and immunotherapeutic strategies in glioblastoma. Specifically, we have replaced references 4, 5, 7, and 8 with the following updated citations:

4, Obrador, E.; Moreno-Murciano, P.; Oriol-Caballo, M.; López-Blanch, R.; Pineda, B.; Gutiérrez-Arroyo, J. L.; Loras, A.; Gonzalez-Bonet, L. G.; Martinez-Cadenas, C.; Estrela, J. M.; et al. Glioblastoma Therapy: Past, Present and Future. 2024, 25(5), 2529.

5, Osurdo, A.; Di Muzio, A.; Cianciotti, B.C.; Dipasquale, A.; Persico, P.; Barigazzi, C.; Bono, B.; Feno, S.; Pessina, F.; Santoro, A.; et al. T Cell Features in Glioblastoma May Guide Therapeutic Strategies to Overcome Microenvironment Immunosuppression. Cancers 2024, 16, 6038.

7, Chen, J.; Wu, Q.; Berglund, A.E.; Macaulay, R.J.; Mulé, J.J.; Etame, A.B. Tumor-Associated Macrophages in Glioblastoma: Mechanisms of Tumor Progression and Therapeutic Strategies. Cells 2025, 14, 1458.

8, Asioli, S.; Gatto, L.; Vardy, U.; Agostinelli, C.; Di Nunno, V.; Righi, S.; Tosoni, A.; Ambrosi, F.; Bartolini, S.; Giannini, C.; et al. Immunophenotypic Profile of Adult Glioblastoma IDH-Wildtype Microenvironment: A Cohort Study. Cancers 2024, 16, 3859.

  1. A minor point is the need to review the uniformity of some acronyms, such as TAMs, where sometimes "Proliferation-TAMs" appears and sometimes "Proliferative TAMs."

Response: We thank the reviewer for this helpful observation. We have carefully reviewed the manuscript and standardized the terminology across all sections. All subtype names have now been unified to the format “Proliferation-TAMs” for consistency.

Round 2

Reviewer 1 Report

Comments and Suggestions for Authors

The authors have made all the changes suggested by the reviewers and I believe that this new and improved version of the manuscript is suitable for publication.

Reviewer 2 Report

Comments and Suggestions for Authors

Authors have addressed the critical concerns.

Reviewer 3 Report

Comments and Suggestions for Authors

I consider that all the requirements pointed out have been properly addressed, with the exception of the font size in the figures, which still does not allow for adequate visualization of the information. Therefore, I believe it is important that this modification be made, adjusting the font size in both the figures of the manuscript and the supplementary material